

# Identification by shape-based virtual screening and evaluation of new tyrosinase inhibitors

Qi Li[1], Hongyu Yang[1], Jun Mo[1], Yao Chen[2], Yue Wu[3], Chen Kang[4], Yuan Sun[5] and Haopeng Sun[1]

[1] Department of Medicinal Chemistry, China Pharmaceutical University, Nanjing, China
[2] School of Pharmacy, Nanjing University of Chinese Medicine, Nanjing, China
[3] Nanjing Duoyuan Biochemistry Co., Ltd., Nanjing, China
[4] Department of Internal Medicine, Carver College of Medicine, University of Iowa, Iowa City, IA, United States of America
[5] Department of Biochemistry and Molecular Medicine, University of California, Davis, Sacramento, CA, United States of America

## ABSTRACT

Targeting tyrosinase is considered to be an effective way to control the production of melanin. Tyrosinase inhibitor is anticipated to provide new therapy to prevent skin pigmentation, melanoma and neurodegenerative diseases. Herein, we report our results in identifying new tyrosinase inhibitors. The shape-based virtual screening was performed to discover new tyrosinase inhibitors. Thirteen potential hits derived from virtual screening were tested by biological determinations. Compound 5186-0429 exhibited the most potent inhibitory activity. It dose-dependently inhibited the activity of tyrosinase, with the $IC_{50}$ values $6.2 \pm 2.0\ \mu M$ and $10.3 \pm 5.4\ \mu M$ on tyrosine and L-Dopa formation, respectively. The kinetic study of 5186-0429 demonstrated that this compound acted as a competitive inhibitor. We believe the discoveries here could serve as a good starting point for further design of potent tyrosinase inhibitor.

# INTRODUCTION

Melanogenesis is a physiological melanin-producing process. The final product of melanogenesis, melanin, which is the pigment produced by epidermis melanocytes, is the main determining factor of pigmentation of human eyes and skin (*Hassan et al., 2017*; *Pillaiyar, Manickam & Jung, 2015*; *Saeed et al., 2017*). Under normal physiological states, melanin is primarily responsible for protecting skin against damaging effects of ultraviolet radiation (*Abbas et al., 2017a*; *Costin & Hearing, 2007*). Besides, melanin is considered to be the main determinant of skin color (*Pillaiyar, Manickam & Namasivayam, 2017*). Melanogenesis is a complex synthetic process involving a series of chemical catalyzed and enzymatic reactions (*Hassan et al., 2016*). Among them, tyrosinase is a well-known enzyme acting as the limiting step in melanogenesis (*Wang et al., 2016*). It catalyzes the hydroxylation of tyrosine (L-Tyr) to form 3,4-dihydroxyphenylalanine (L-Dopa), which can be subsequently oxidized to form *o*-dopaquinone (*Ferro et al., 2017*). Further oxidation

Corresponding author
Haopeng Sun,
sunhaopeng@cpu.edu.cn,
sunhaopeng@163.com

and reduction steps lead to the production of melanin pigments, such as pheomelanin and eumelanin (*Khatib et al., 2007*). Although the melanin serves as a major defense mechanism against UV light in human skin, excessive melanin production and accumulation of epidermal pigmentation can cause various hyperpigmentation disorders, including senile lentigines, freckles, ephelide and melisma (*Abbas et al., 2017b*; *Solano et al., 2006*; *Unver et al., 2006*). Moreover, it was recently demonstrated that tyrosinase is involved in the biosynthesis of neuromelanin in nigrostriatal dopamine neurons of the central nervous system (CNS) (*Vontzalidou et al., 2012*). Overexpression of tyrosinase leads to the massive oxidation of L-Dopa and dopamine into dopaquinone and dopamine quinone, both of which are found to result in neuronal damage and cell death, linking tyrosinase to neurodegenerative diseases, especially Parkinson's disease (PD) (*Tessari et al., 2008*).

Due to the critical role of tyrosinase in the melanogenesis process, it becomes an attractive target for medicinal chemists to develop inhibitors applied in cosmetics and pharmacological therapy (*Larik et al., 2017*). Recently, different research groups have engaged in the discovery of tyrosinase inhibitors, and several new types of molecules were disclosed (*Ashraf et al., 2015*; *Choi et al., 2014*; *Haudecoeur et al., 2017*; *Mojzych et al., 2017*; *Park et al., 2013*; *Ruzza et al., 2016*; *Tan et al., 2016*; *Wang et al., 2016*; *Wu et al., 2017*; *You et al., 2015*). These findings provide opportunity for the subsequent medicinal optimization. However, these active compounds, many of which are derived from natural products, are usually limited by low activity, poor target selectivity, unsatisfied physicochemical properties, or difficulty in availability. Therefore, identification of new tyrosinase inhibitors with synthetic scaffold is an attractive and challenging task for medicinal chemists. Herein, we disclose our recent findings in the identification of new small molecule tyrosinase inhibitors by using the ensemble-based virtual screening method.

## EXPERIMENTAL METHODS

### Creation of the ROCS overlays and the validation

Shape-based overlays used in further virtual screening were generated by using Rapid Overlay of Chemical Structures (ROCS). Neorauflavane was used as template molecule (*Tan et al., 2016*). The underlying idea of the algorithm is that different compounds could share similar molecular shapes if they can overlay well and any mismatch in volume is from the dissimilarity in shapes (*Kirchmair et al., 2007*). ROCS maximizes the rigid body overlap of the molecular Gaussian functions and resulting in the shared volume between a query molecule and a conformation of a database molecule (*Zhou et al., 2008*). A smooth Gaussian function is employed to determine the molecular volume and serve for the superimposition of molecules. Following that, chemical functionalities are simply aligned and matched for the overlay of molecules. Briefly, the ROCS color force field assigns six types of chemical features such as hydrogen bond acceptor, hydrogen bond donor, hydrophobes, cations, anions and rings to one molecule by spatial arrangement (*Xue et al., 2016*). The combo score ranges from 0~2, the higher the score is, the more similar of a given compound is to neorauflavane.

## Virtual screening of the database

Virtual screenings were then performed on the basis of the query overlay of neorauflavane using ROCS. The following parameters were set for the run of ROCS: rankby = combo and besthits = 1. During the screening, ROCS compares database compounds and the control molecular by aligning the compounds and calculating the similarities including their volumes and chemical features. The similarity is evaluated and represented by a combo score, ranging from 0 to 2. With combo score close to 2, molecules have an excellent shape and chemical-feature match, while value close to 0 implies poor shape and chemical-feature similarities. Chemdiv and Specs compound collections with 315,000 compounds were screened by this model (*Kirchmair et al., 2009*). Finally, 13 compounds were retained and purchased from Topscience database with purity >95% (Liquid chromatography–mass spectrometry, LC-MS).

## *In vitro* tyrosinase inhibition assay

The assay followed the method of *Masamoto et al. (2003)* with slight modifications. Briefly, mushroom tyrosinase (T3824), L-tyrosine (91515), L-Dopa (D9628) were purchased from Sigma-Aldrich (Shanghai, China). (It is worth noting that, as no human tyrosinase can be obtained from commercial resources, we here used mushroom tyrosinase as a substitution, like many excellent studies do (*Ashraf et al., 2015*; *Larik et al., 2017*; *Masamoto et al., 2003*; *Saeed et al., 2017*). Although the entire structure of mushroom tyrosinase is different from human-origin tyrosinase, the catalytic sites of both two kinds of tyrosinase are very similar.) Aliquots (5 µL) of test compounds at various concentrations ($10^{-11} \sim 10^{-4}$ M, dissolved in methanol and diluted by water) were mixed with 50 µL of L-tyrosine or L-Dopa solution (1.25 mM, prepared in water) respectively, 90 µL of sodium phosphate buffer solution (0.05 M, pH 6.8) and preincubated at 25 °C for 10 min. Then a 5 µL aqueous solution of mushroom tyrosinase (333 U/mL) was added into the mixture. The absorbance at 475 nm was measured after 30 or 5 min of incubation time of the reaction mixture containing L-tyrosine or L-Dopa, respectively (*Masamoto, Lida & Kubo, 1980*) The inhibitory activity of samples is expressed as inhibition percentage and calculated as follows: Inhibition % = $\{[(A-B)-(C-D)]/A-B\} * 100$ (A: Abs of phosphate buffer and enzyme; B: Abs of phosphate buffer; C: Abs of phosphate buffer, test sample and enzyme; D: Abs of phosphate buffer and test sample.). All inhibitory experiments were carried out in parallel triplicate and averaged. Calculation of the $IC_{50}$ values was performed with Graph Pad Prism 5.0. Kojic acid was used as standard inhibitors for this experiment.

## Kinetic study

Kinetic measurements were performed in the same manner, while the substrate L-Dopa was used in concentrations of 39.1, 78.2, 156.3, 312.5, and 625 µM for each test compound concentration (0, 1.25, 2.5, 5, and 10 µM). The enzymatic reaction was set to 7 min before the absorption was measured. $V_{max}$ and $K_m$ values (for Michaelis–Menten kinetics) were obtained with Graph Pad Prism 5.0 from the nonlinear regression of substrate-velocity curves (Table 1). Linear regression was obtained as Lineweaver-Burk plots of 1/V versus 1/[S], giving diverse slopes (*You et al., 2015*).

**Table 1** $V_{max}$ and $K_m$ values for each test compound concentration.

| Concentration, $\mu$M | $V_{max}$, $\mu$M min$^{-1}$ mg$^{-1}$ | $K_m$, $\mu$M |
|---|---|---|
| 10 | 49.16 | 114.8 |
| 5 | 46.53 | 195.2 |
| 2.5 | 51.31 | 223.8 |
| 1.25 | 45.09 | 231.9 |
| 0 | 56.15 | 232.9 |

## Cell viability

Cell viability was quantified by colorimetric MTT assay by measuring mitochondrial activity in living cells. 3-(4,5-Dimethylthiazol-2-yl)-2,5-diphenyltetrazoliumbromide (MTT; Sigma) was used to evaluate cytotoxicity of **5186-0429** on B16F10 melanoma cells. MTT could be transformed into MTT-formazan crystal by mitochondrial enzyme (*Tada et al., 1986*). Cells were firstly seeded into a 96-well culture plates at a density of $5 \times 10^4$. **5186-0429** was diluted with DMEM containing 1% fetal bovine serum (FBS; Gibco, Carlsbad, CA) and applied to the 96-well culture plates with B16F10 cells. Cells were allowed to adhere after 24 h. Then the culture medium was replaced by fresh serum-free DMEM. MTT was prepared at 5 mg/ml in phosphate-buffered saline (PBS). 500 $\mu$l of MTT stock solution were added into each well, and the plate was incubated at 37 °C for 4 h. After that, PBS was removed. To each well, 500 $\mu$L of EtOH-DMSO (V : V = 1:1 mixture solution) was added to dissolve formazan (*Park et al., 2013*). After 10 min, the absorbance of the colored solution was carried out spectrophotometrically with a wavelength of 560 nm filter.

## Molecular docking

The molecular docking study was conducted in Discovery Studio 3.0 with CDOCKER module. The docking procedures are summarized as following. To begin with, ligand "seeds" were created by CDOCKER and used to occupy the binding pocket. Molecular dynamics (MD) with high temperature is then applied to each generated seed by using a modified CHARMm force field. After that, the aquired structures are completely minimized under the same forcefield. Furthermore, based on the positions and conformations, the solutions are clustered and ranked by energy. The crystal structure of Agaricus Bisporus mushroom tyrosinase bound with tropolone (PDB id: 2Y9X) was adopted for docking study. Residues around the tropolone of tyrosinase (radius = 6 Å) were chosen to construct the binding sites. The heating and cooling steps were both set to 5000, while the temperature was kept at 310. Default setting were used for other parameters.

## RESULTS

### Creation of the ROCS overlays and the validation

The virtual screening was initiated by a shape-based screening by using ROCS methods. ROCS is a well-known virtual screening program using shape comparison application and gaussian scoring function (*Kirchmair et al., 2009*). After searching the recently published references, a natural product, neorauflavane (*Tan et al., 2016*) (Fig. 1A), was selected as the reference compound when generating the shape-based screening model. As neorauflavane

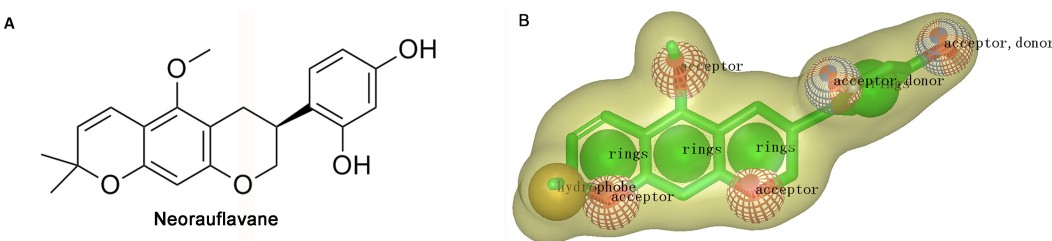

**Figure 1** (A) The structure of neorauflavane. (B) The shape-based screening model for the identification of new tyrosinase inhibitors.

was one of the most active tyrosinase inhibitors published in the references ($IC_{50}$ values of $0.03 \pm 0.006\ \mu M$ and $0.5 \pm 0.03\ \mu M$ on L-Tyr and L-Dopa, respectively), we concluded that screening hits with similar molecular shape to neorauflavane may have good tyrosinase inhibitory effects.

The shape-based model was shown in Fig. 1B. The shape of neorauflavane was displayed in light yellow shadow. The tricyclic core of neorauflavane was defined by three adjacent ring features. The two oxygen atoms on the tricyclic core, and the oxygen atom of methoxyl group, provided three hydrogen-bond acceptor features. The gem-dimethyl group served as a hydrophobe. The resorcinol moiety supplied one ring feature, two hydrogen-bond acceptor features and two hydrogen-bond donor features.

## Virtual screening of the database and *in vitro* tyrosinase Inhibition Assay

The shape-based model was applied in the virtual screening of commercial compound libraries including Chemdiv and Specs with a collection of 315,000 compounds. The similarity in molecular shape between the screened compounds and neorauflavane was evaluated by the combo score method, which consisted of the shape Tanimoto coefficient and the score retrieved from the ROCS color force field, which stand for the structural complementarity between the template and the screened molecules. Finally, 13 compounds (Fig. 2) were purchased from Topscience cooperation and initially screened for their tyrosinase inhibitory rate (IR) under the concentration of $10\ \mu M$ (*Ferro et al., 2017*; *You et al., 2015*). Kojic acid was used as positive control. Among them, three compounds, **3253-1775**, **5186-0429** and **3720-3263**, exhibited over 40% inhibitory efficiency on L-Tyr oxidation, while only **5186-0429** showed IR over 40% on L-Dopa oxidation (Figs. 3A & 3B). Besides, **5186-0429** was the most potent compound in the initial screen (Table 2); therefore, it was further evaluated for the $IC_{50}$ value. According to the results, **5186-0429** dose-dependently inhibited the activity of tyrosinase (Fig. 3C), with $IC_{50}$ values of $6.2 \pm 2.0\ \mu M$ ($R^2 = 0.9711$) and $10.3 \pm 5.4\ \mu M$ ($R^2 = 0.9589$) on L-Tyr and L-Dopa, respectively (Table 2). These data suggested that **5186-0429** can serve as a positive hit for further development of active tyrosinase inhibitors.

**Figure 2 Hit compounds purchased for tyrosinase inhibition assay.** Each structure of the 13 compounds was shown in A–M, respectively.

## Kinetic study

To illuminate the inhibition mechanism of **5186-0429** on tyrosinase, the kinetic study, with L-Dopa as the substrate, was carried out by using Lineweaver-Burk plots, which characterize the reciprocal rates versus reciprocal substrate concentrations under different inhibitor concentrations resulting from the substrate-velocity curves for tyrosinase. Under different concentrations of **5186-0429** (lines in Fig. 3D), the intersection located on the vertical axis, suggesting that this compound competed with L-Dopa when it bound to tyrosinase. Therefore, we speculated that **5186-0429** could enter the active center of tyrosinase, thereby acting as a competitive inhibitor, with a $K_i$ value of 12.2 μM.

## Cell viability

To evaluate the cytotoxic effect of **5186-0429**, we used the murine B16F10 melanoma cell line (B16F10 cells). The results of cell viability assay using an MTT kit are presented in Fig. 4. At the doses of 1, 5, 10, 50, 100 μM of compound **5186-0429**, cell viability results were 100%, 96.17%, 81.68%, 5.17%, 5.26%, respectively. It indicated that compound **5186-0429** is not cytotoxic to B16F10 cells at low concentrations.

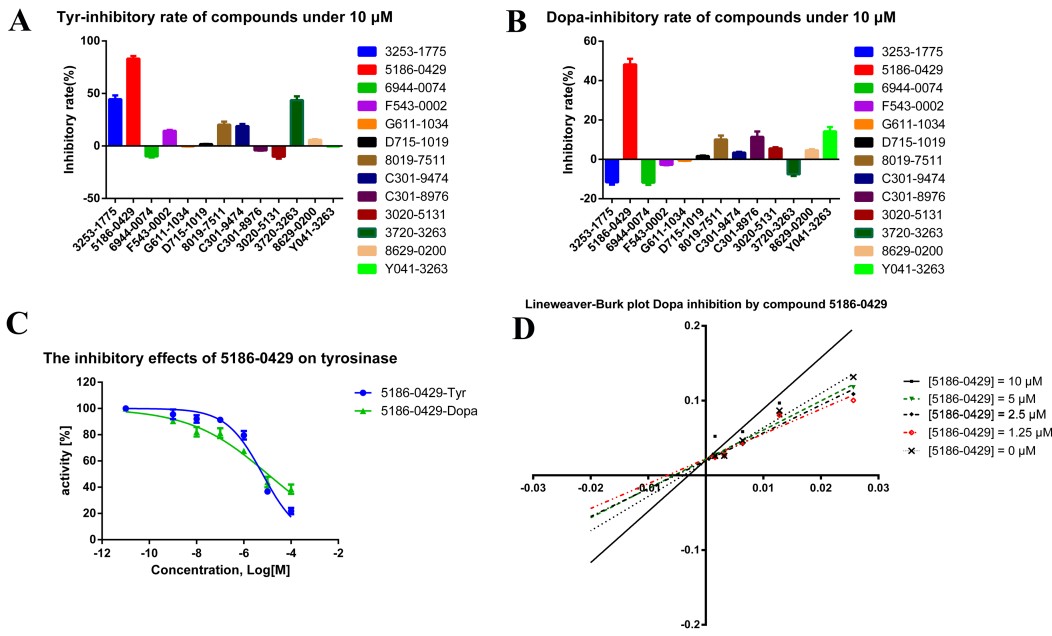

**Figure 3** **The assays for tyrosinase inhibitory activity of the hits.** (A, B) The initial evaluation of the 13 hits on tyrosinase at 10 $\mu$M. (C) The dose-dependent inhibitory manner of the most potent hit 5186-0429 on L-Tyr and L-Dopa. (D) The Lineweaver-burk plot of 5186-0429 in kinetics assays.

**Table 2** **The initial assay of the 13 hits on the tyrosinase inhibition.**

| Cpd. | Tyr IR (%)[a] | Dopa IR (%)[a] |
|---|---|---|
| **3253-1775** | 44.56 | −11.63 |
| **5186-0429** | 83.19 $6.2 \pm 2.0 \, \mu M^{b}$ | 48.23 $10.3 \pm 5.4 \, \mu M^{b}$ |
| **6944-0074** | −9.96 | −11.83 |
| **F543-0002** | 14.45 | −2.75 |
| **G611-1034** | −0.46 | −0.68 |
| **D715-1019** | 1.86 | 1.71 |
| **8019-7511** | 20.30 | 10.08 |
| **C301-9474** | 18.89 | 3.41 |
| **C301-8976** | −4.17 | 11.42 |
| **8020-5131** | −10.19 | 5.54 |
| **3720-3263** | 43.74 | −7.54 |
| **8629-0200** | 5.87 | 4.65 |
| **Y041-3263** | −0.52 | 14.25 |

**Notes.**
[a] The inhibitory rate of Tyr or Dopa by the compounds.
[b] The $IC_{50}$ values of Tyr or Dopa by **5186-0429**.

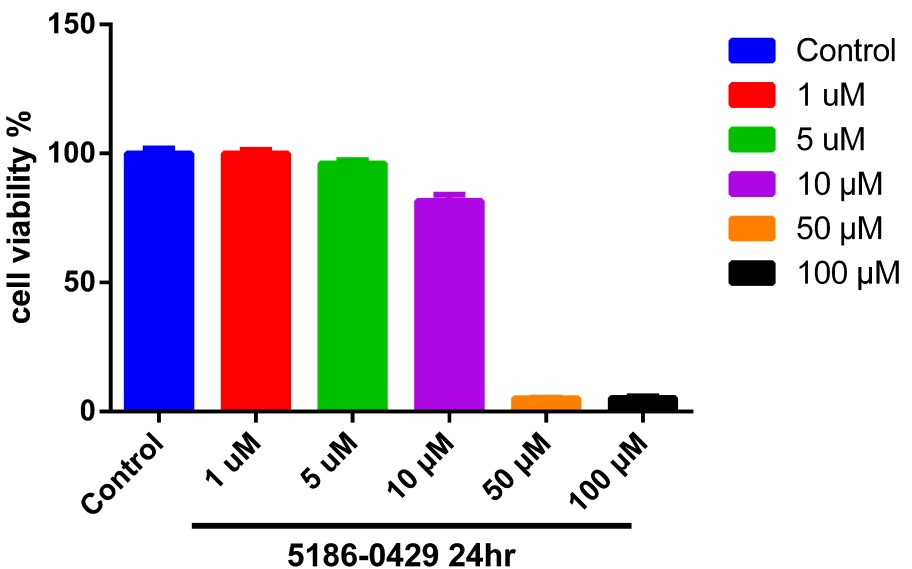

**Figure 4 Effect of 5186-0429 on B16F10 cell viability.** Cells were treated with diverse doses of 5186-0429 (1–100 μM) for 24 h and evaluated by an MTT assay. Data are expressed as a percentage of the control group.

## Molecular docking

Molecular docking was applied to further analyze the binding mode between **5186-0429** and tyrosinase. The structure of mushroom tyrosinase was downloaded from protein data bank (PDB ID: 2Y9X). The docking was performed using CDOCKER module in BIOVIA Discovery Studio (DS). The docking pose selection was on the basis of its best CDOCKER energy (−22.7987 kJ) and CDOCKER interaction energy (−35.6582 KJ). The According to the results (Fig. 5), the resorcinol moiety played a central role during the intermolecular interaction. One hydroxyl established a hydrogen-bond with the sidechain of Asn260, while the other hydroxyl interacted with the copper ion, which was critical for the enzymatic activity of tyrosinase. The benzene ring of resorcinol moiety formed π–π, π-alkyl and π-sigma interaction with His263, Ser282 and Val283, respectively. The thiazole moiety of **5186-0429** formed a π–π interaction with the sidechain of Phe264. Additionally, a π-sulfur interaction was observed between the sulfur atom of **5186-0429** and the sidechain of Phe264. For the p-phenylenediamine moiety of **5186-0429**, it formed π–π and π-alkyl interaction with Phe264 and Val248, respectively. Finally, a series of residues, including His61, His85, His244, Glu256, Met257, His259, Arg268, Ala286 and Phe292, were involved in the intermolecular recognition with **5186-0429** through van der Waals interactions, which further improved the binding affinity of this compound.

## DISCUSSION

Tyrosinase is a limiting enzyme in melanogenesis process. Besides, it is involved in the biosynthesis of neuromelanin in nigrostriatal dopamine neurons of CNS. Its overexpression results in neuronal damage and cell death, which are characterized in
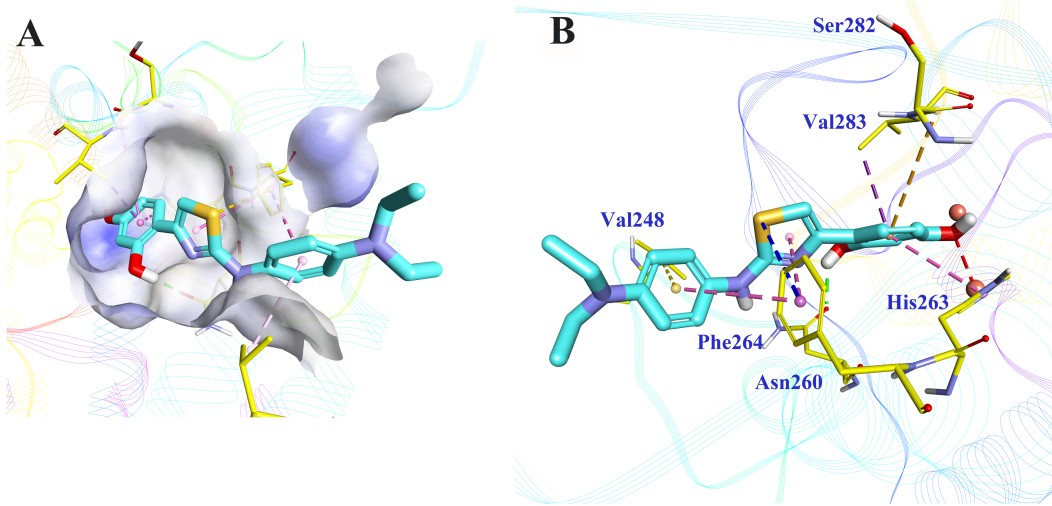

**Figure 5  The binding mode prediction of 5186-0429 with tyrosinase.** Binding pocket and ligand conformation were exhibited (A). The binding pattern was shown in a three-dimensional presentation (B). Dotted lines in different colors reflected diverse interaction types (hydrogen bonds in green, metal-acceptor interaction in grey, Pi-sigma in purple, Pi-sulfur in blue, Pi-Pi interaction in pink, amide-Pi in orange and Pi-alkyl in yellow).

several neurodegenerative diseases, especially PD. Thus, inhibition of tyrosinase is of vital importance.

Although diverse researches have disclosed several types of tyrosinase inhibitors, most of them are derived from natural products. Limitations, like low activity and selectivity, poor availability and novelty, unsatisfied physicochemical properties, etc., cannot be ignored. Herein, we engaged in virtual screening to identify novel tyrosinase inhibitors, with relatively small molecular weight but higher activity and lower cytotoxicity.

As a result, **5186-0429** exhibited over 40% inhibitory efficiency on L-Tyr oxidation and IR over 40% on L-Dopa oxidation, with a considerable $IC_{50}$ values of $6.2 \pm 2.0\,\mu M$ and $10.3 \pm 5.4\,\mu M$ on L-Tyr and L-Dopa, respectively. Kinetic study showed that **5186-0429** worked as a competitive inhibitor with $K_i$ value of $12.2\,\mu M$. That may be because it could enter and act at the active center of tyrosinase. Due to cell viability results were 100%, 96.17%, 81.68% at doses of 1, 5, 10 $\mu M$ respectively, compound **5186-0429** showed no cytotoxicity to B16F19 cells at low concentrations. Hydrogen bond between one hydroxyl and Asn260 and interaction of the other hydroxyl with copper ion are prerequisite for enzymatic activity of tyrosinase. Meanwhile, several π–π, π-alkyl, π-sulfur and π-sigma interactions enhanced the inhibition. Intermolecular van der Waals interactions of **5186-0429** with surrounding amino acid residues greatly improved the binding affinity. The distance between Cu ion and methoxyl group is 2.629 Å.

Compound **5186-0429** was identified via virtual screening, with micromolar grade of inhibition activity and no cytotoxicity at low concentration. Discovered from neither structure modification nor plant extraction, **5186-0429** possesses obvious structural novelty. However, $IC_{50}$ values of **5186-0429** are only with micromolar grade both on

L-Tyr and L-Dopa. Much effort should be taken to enhance the inhibitory ability. As described before, the binding mode of **5186-0429** and its target tyrosinase was disclosed, which could provide a modification template for further design and development. For example, cavity around resorcinol seems possible to accept a larger moiety. Bi- or tricyclic moieties may be more proper to fit this binding pocket. Besides, as the distance between Cu ion and methoxyl group is disclosed, an atom with lone pair electrons is essential to form a coordinate covalent bond with Cu ion. In addition, N,N-diethyl moiety stretches into solvent without forming key interactions with the receptor. Thus, this group is possible to be modified in order to enhance pharmacokinetic properties. From a clinical perspective, tyrosinase inhibitors with high solubility can be developed into eye-drop preparations to cure ophthalmic diseases caused by melanin deposits. In our subsequent work, more modifications and designs are still in need. With the abundance of molecule structures, structure–activity relationship will be carried out, inhibitors with higher activity and selectivity will be discovered, and how this series of compounds work will certainly be clearer.

## CONCLUSIONS

In conclusion, a new tyrosinase inhibitor, **5186-0429**, was discovered through shape-based virtual screening with the template of neorauflavane. **5186-0429**, with micromolar inhibitory activity, inhibited tyrosinase in a competitive manner according to the kinetic and molecular docking study. All above suggested **5186-0429** to be a potential hit for further modification and development as a novel tyrosinase inhibitor. It provides a good template for further design of a potent tyrosinase inhibitor.

### Funding

This work was supported by grants 81402851 and 81573281 from the National Natural Science Foundation of China, and BK20140957 from the Natural Science Foundation of Jiangsu Province. Support was also recieved from Fundamental Research Funds for the Central Universities (2015ZD009), Jiangsu Qing Lan Project, Top-notch Academic Programs Project of Jiangsu Higher Education Institutions (TAPP-PPZY2015A070) and Priority Academic Program Development of Jiangsu Higher Education Institutions (PAPD). The funders had no role in study design, data collection and analysis, decision to publish, or preparation of the manuscript.

### Grant Disclosures

The following grant information was disclosed by the authors:
National Natural Science Foundation of China: 81402851, 81573281.
Natural Science Foundation of Jiangsu Province: BK20140957.
Fundamental Research Funds: 2015ZD009.
Jiangsu Higher Education Institutions: TAPP-PPZY2015A070.
Jiangsu Higher Education Institutions (PAPD).

## Competing Interests

The authors declare there are no competing interests.

## Author Contributions

- Qi Li, Hongyu Yang, Jun Mo, Yao Chen, Yue Wu and Haopeng Sun conceived and designed the experiments, performed the experiments, analyzed the data, contributed reagents/materials/analysis tools, wrote the paper, prepared figures and/or tables, reviewed drafts of the paper.
- Chen Kang and Yuan Sun conceived and designed the experiments, wrote the paper, prepared figures and/or tables, reviewed drafts of the paper.

## Data Availability

The raw data is provided in the Supplemental Files.

## Supplemental Information

Supplemental information for this article can be found online at http://dx.doi.org/10.7717/peerj.4206#supplemental-information.

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
