# Peer review of "Identification by shape-based virtual screening and evaluation of new tyrosinase inhibitors"

_PeerJ, doi:10.7717/peerj.4206_

## Round 0.1 · original submission · Minor Revisions

· Academic Editor

Minor Revisions

My personal opinion is that the manuscript deserves publication on PeerJ provided some additional information and a better contextualization of the work done and the results achieved are given.

In particular:

Modelling. 1) Virtual screening: some details should be added, like, for example, which (name of the library) and how many compounds were screened; which were the criteria that led to the selection of those 13 compounds reported. 2) Docking. A comparison of the docking results for compound 5186-0429 with those for the reference compound neorauflavane, obtained by you and reported in the literature (Tan Bioorg Med Chem 2016), has to be provided. My opinion is that you should concentrate more on describing your results, rather than making additional work as suggested by reviewer 1, like performing additional screening using “Kojic acid and Arbutine as a template for screening”, which is out of the scope of this manuscript.

Biological evaluation: reviewer 1 asks for IC50 values for all the 13 inhibitors. I’m not sure that this would be possible as I suppose many of them were inactive/very poorly active. However, add a sentence describing and discussing their behavior. As for the other requests made by the reviewers on the biological part, please try to answer to all of them. In particular the issue of riproducubility of experiments raised by reviewer 2 has to be taken very seriously.

Moreover, for those not experts in the field, you should give an idea on the transferability of your results obtained using mushroom tyrosinase to human tyrosinase.

Please provide a resubmission letter in which you answer point by point to all the observations of the reviewers and to mine and in which you describe the changes made to the text of the manuscript.

Best regards,
Silvia Rivara.

·

Basic reporting

The overall data in manuscript is interesting and informative.

Experimental design

Good experimental approach was used in manuscript

Validity of the findings

Authors tried to verify their results using various approaches. However, further techniques are required to validate their results.

Additional comments

Comments
The present manuscript screen the possible inhibitors of mushroom tyrosinase. The manuscript is well written and the findings are interesting. Therefore, article can be consider for publication after revision.

• The manuscript title “Identification of new tyrosinase inhibitors with shape-based virtual screening” is just emphasize on in-silico based study, whereas authors also performed in-vitro experiments. Therefore, manuscript title must be rephrased. Moreover, add few studies such as (Eur. J. Med. Chem. 141: 273-281; DOI: 10.1007/s12539-016-0171-x; Comput Biol Chem. 2017 Jun; 68:131-142; Chem Biodivers. 2017 Sep;14(9); Drug Des Devel Ther, 2017, 11:2029-2046; Bioorg Chem, 2017, 74:187-196) in introduction part.
• Authors used Neorauflavane as a template, is it possible to use Kojic acid and Arbutine as a template for screening as you used in in-vitro experiment as a positive control.
• Author did not discuss about size of protein structure, domain and binding pocket. I would suggest to make one graphical image of all inhibitors or your potent inhibitor against tyrosinase to show binding pocket and ligand conformation.
• Authors did not discuss how many conformations he selected in docking experiments and on what basis he selected best pose of among all either just energy value or binding interaction pattern. In docking figure authors did not mentioned the binding distances of interactions. Moreover, authors claims “One hydroxyl established a hydrogen bond with the side chain of Asn260, while the other hydroxyl interacted with the copper ion” in docking results. However, it’s not clear from graphics either it is hydrogen bond or some other type of interaction. In discovery studio mostly the hydrogen bonds are represented by green dotted lines. Moreover, authors did not mentioned docking energy value for 5186-0429 in the results part.
• Moreover, I would suggest (if possible) to run MD simulation using any software to check the stability of target protein.
• For in-vitro analysis Table 1, Authors only mentioned IC50 value for 5186-0429, however, it’s better to calculate IC50 values for all other compounds in μM. It would help in comparison amongst all screened compounds with control.
• In MM section, authors mentioned “Vmax and Km values (for Michaelis-Menten kinetics) were obtained with Graph Pad Prism 5.0 from the nonlinear regression of substrate-velocity curves”. However we could not found Vmax and Km values in manuscript file. Moreover, also provide Ki value (Ki is the EI dissociation constant).
• Authors used two substrate (L-Tyro and L-Dopa) for inhibition purpose as mentioned in MM section, while in kinetic study only L-Dopa was used to generate Lineweaver-Burk plots. Therefore, I would suggest to use both substrates and make separate graphs for both substrate.
• Furthermore, I would suggest (if possible) to use Zebrafish model for melanin quantification along with toxicity assay for your most active compound.

Reviewer 2 ·

Basic reporting

No comment

Experimental design

*In which solvent the L-DOPA, Tyrosine and test samples were prepared?

"0.9 mL of sodium phosphate buffer solution (0.05 M, pH 6.8)" This volume of PBS was fixed for all samples? If this is true, we will have difference in the concentrations after the addition of the test samples.

Validity of the findings

Some of the data are without error bars. For example: Fig. 3A-B and Fig. 4. How about the reproducibility of these experiments? At least two independent experiments, preferably three with error bars, are required.

In graph 3C determine the values of R-saquared and P value and Calculate the values of Ki,

The major point is: the discussion of the results needs to be reviewed, relating the theoretical data with in vitro

Additional comments

The design of the study appears appropriate and most of the data are satisfactory. However, there are some points that need to be improved, as summarized above.

---

## Round 0.2 · accepted · Accept

· Academic Editor

Accept

My opinion is that the manuscript by Li et al. (#19887) should be accepted for publication. The authors answered to the questions of the reviewers and editor and improved the text of the manuscript when possible.

I suggest the authors to make some minor changes in their last revised form of the manuscript:

1) Title: It is not correct “... bio-evaluation ...with shape-based virtual screening”. I would suggest “Identification by shape-based virtual screening and evaluation of new tyrosinase inhibitors”.
2) Abstract: “The kinetic study and molecular docking of 5186-0429 demonstrated that this compound acted as a competitive inhibitor.” Delete molecular modelling from the sentence as it demonstrated nothing about the kinetics of inhibition.
3) line 137: force field; line 166: Tanimoto
4) Figure 1: flip figure 1A vertically to show the molecule as it is represented in Figure 1B.
5) Figure 3: The assays for tyrosinase inhibitory activity of the hits. (A, B) The initially evaluation of the 13 hits on tyrosinase at 10 μM.
6) line 233: we engaged in virtual screening to identify novel tyrosinase inhibitors; the same (develop/identify) at line 246
7) when reporting Ki value for the best compound use only significant digits.
8) line 257: pharmacokinetic properties

Please re-check English language carefully.